# Insights into Bioinformatic Applications for Glycosylation: Instigating an Awakening towards Applying Glycoinformatic Resources for Cancer Diagnosis and Therapy

**DOI:** 10.3390/ijms21249336

**Published:** 2020-12-08

**Authors:** Manikandan Muthu, Sechul Chun, Judy Gopal, Vimala Anthonydhason, Steve W. Haga, Anna Jacintha Prameela Devadoss, Jae-Wook Oh

**Affiliations:** 1Department of Environmental Health Sciences, Konkuk University, Seoul 143-701, Korea; bhagatmani@gmail.com (M.M.); scchun@konkuk.ac.kr (S.C.); jejudy777@gmail.com (J.G.); awesomeannaa1@gmail.com (A.J.P.D.); 2Department of Microbiology and Immunology, Institute for Biomedicine, Gothenburg University, 413 90 Gothenburg, Sweden; vimalalisha@gmail.com; 3Department of Computer Science and Engineering, National Sun Yat Sen University, Kaohsiung 804, Taiwan; stevewhaga@yahoo.com; 4Department of Stem Cell and Regenerative Biotechnology, Konkuk University, Seoul 143-701, Korea

**Keywords:** glycosylation, cancer, bioinformatics tools, databases, post translational modification, proteins

## Abstract

Glycosylation plays a crucial role in various diseases and their etiology. This has led to a clear understanding on the functions of carbohydrates in cell communication, which eventually will result in novel therapeutic approaches for treatment of various disease. Glycomics has now become one among the top ten technologies that will change the future. The direct implication of glycosylation as a hallmark of cancer and for cancer therapy is well established. As in proteomics, where bioinformatics tools have led to revolutionary achievements, bioinformatics resources for glycosylation have improved its practical implication. Bioinformatics tools, algorithms and databases are a mandatory requirement to manage and successfully analyze large amount of glycobiological data generated from glycosylation studies. This review consolidates all the available tools and their applications in glycosylation research. The achievements made through the use of bioinformatics into glycosylation studies are also presented. The importance of glycosylation in cancer diagnosis and therapy is discussed and the gap in the application of widely available glyco-informatic tools for cancer research is highlighted. This review is expected to bring an awakening amongst glyco-informaticians as well as cancer biologists to bridge this gap, to exploit the available glyco-informatic tools for cancer.

## 1. Introduction

Post translational modification (PTM) is the modification of proteins subsequent to protein biosynthesis. Proteins, undergo PTM to form the anticipated protein product. There are various forms of PTMs, namely hydroxylation, glycosylation, phosphorylation, methylation. ubiquitination, SUMOylation, acetylation and lipidation (Figure 1). Glycosylation is associated with a nework of critical biological processes. The process involves attachment of carbohydrate molecules to lipids and protein moieties. Monosaccharides that are covalently linked by glycosidic bonds, are called glycans. N-glycosylation and O-glycosylation are the two major glycosylation types. C-glycosylation (or C-mannosylation), glypiation (or glycosylphosphatidylinositol (GPI) anchoring) and glycation (or nonenzymatic glycosylation) also exist [1,2].

The biological roles of glycosylation include diverse functions and influence protein folding and oligomerization [3], protein degradation [4], epitope recognition [5], protein solubility and stability [6], cell-cell interactions [7] and protein transport [8]. Subsequently, glycoproteins also have a say in cancer development and progression [9], autoimmune diseases [10] and congenital disorders [11]. Tailoring glycosylation for manufacturing biotherapeutics, enhances the safety and efficacy of monoclonal antibodies and erythropoietin. Among all PTMs, glycosylation is established to be the most complex [12]. Biologically active glycan structures are encoded indirectly into the genome and glycosidases, glycosyltransferase and carbohydrate-modifying enzymes help attaching glycan structures to proteins and lipids. [12]. Then the glycoproteins are modified with different carbohydrates resulting in several glycoforms. The recognition of glycan-occupied sites is expensive and laborious and can only be experimentally determined [13]. This is the reason why still the number of verified glycosylated residues are limited as compared to the existing knowledge of the known protein sequences [14]. This knowledge regarding glycosylation site locations is by itself a valuable asset. This knowledge is useful in improving the 3D protein structure prediction, to manipulate protein pharmacokinetic properties through glycoengineering [15]. In this direction, development of bioinformatics tools to predict glycosylation sites gain paramount significance.

Munkley et al. in their review have discussed that aberrant protein glycosylation can be linked with cancer and protein glycoforms as cancer biomarkers. Aberrant glycosylation can have a say in cancer progression, as glycans do on cell adhesion, migration, immune surveillance, interactions with cell matrix, signaling and cell metabolism [16]. Development and progression of cancer is known to result in fundamental changes to the glycome. Aberrant glycosylation has been reported in prostate cancer, where aberrations in glycan composition of prostate cancer cells are linked to disease progression. Glycan variants have been reported to be associated with breast, colon, liver, skin, ovarian and bladder cancers and neurodegenerative diseases [17,18,19]. Many glycoproteins show alterations of glycosylation during cancer [20], these can act as glycan biomarkers for various cancer types. MUC-1 (CA15-3/CA27.29) [21] and plasminogen activator inhibitor (PAI-1) [22] have been reported as biomarkers for beta-human chorionic gonadotropin (Beta-hCG); breast cancer; [23] testicular cancer; alpha-fetoprotein (AFP) [24] for liver cancer and germ cell tumors. Chromogranin A (CgA) [25] is a biomarker for neuroendocrine tumors and biomarkers MUC16 (CA-125) [26] and HE4 [27] for ovarian cancer. The hallmarks of glycosylation in cancer are presented in Figure 2.

Bioinformatics resources for glycomics and glycobiology-related databases has been reviewed [28,29,30,31,32,33,34]. The application of bioinformatic tools in biology has led to a fundamental understanding of the biological processes involved [35,36]. Besides the storage, analysis, annotation, organization, systematization needs of genomics, proteomics, and metabolomics data have been met with the emergence and development of bioinformatics. Bioinformatics has provided glycobiologists with databases and tools to support their research. High throughput analysis of glycans requires automated analysis pipelines that bank on extensive bioinformatic support. Various bioinformatic groups are developing mathematical or statistical algorithms and computational methodologies to analyze biological data. Although bioinformatics for glycobiology or glycomics (“glyco-informatics”) [32] is yet to be established compared to genomics and proteomics [37,38], a steady increase in both the development, and use of glyco-informatics tools and databases has been evidenced [28,29,39,40,41,42,43,44,45,46,47].

This is the first review to look at the use of glycosylation bioinformatics data in cancer research and diagnostics. This review focusses on highlighting the importance of glycosylation in cancer. The achievements made using bioinformatics in the area of glycosylation are reviewed and presented. The scarce inputs available from the use of bioinformatics towards glycosylation with respect to cancer is reviewed for the first time. The future perspective of this integrated approach has been speculated.

## 2. Snapshot of Bioinformatic Resources for Glycosylation

Bioinformatic resources have become popular in glycan research for automation of analytical outputs and for high throughput processing of data. The analysis of glycans is challenging, this is why glycomics is lagging behind genomic and proteomic research. The robotic N-glycan release and labelling combined with UPLC/HPLC analytical platforms with other bioinformatics tools have enabled high-throughput glycoprofiling. High resolution NMR spectroscopy and mass spectrometry have been reported for glycoprofiling [48]. The intervention of bioinformatics resources in glycan research has impacted the successful characterization of biological samples and resulted in progress in biomarker discovery, and genome-wide association studies. Automation technology is a valuable asset when it comes to analysis, when combined with bioinformatics several orders of magnitude higher high throughput processing has been reported [43]. Figure 3 presents the workflow of glycosylation, beginning from wet lab analysis to analytical instrumentation to computation (bioinformatics).

Bioinformatics offers a variety of databases to the glycoscientist [30,39,42,49,50,51]. Glycan informatics has provided varied methods for glycosylation analysis, glycomics and glycan biomarker prediction. Tools are there for prediction of glycosylation binding sites, for statistical analysis of amino acids of a glycoprotein and mathematical modeling of glycosylation. Tools for annotation of mass spectra and for glycan structure analysis and Glyco-Gene Expression Analysis are available for users. Bioinformatics resources for glycoresearch can be grouped as: (i) databases containing information on proteins, (ii) databases storing information on enzymes and pathways building glycans, and (iii) carbohydrate structural databases [42,43]. Not many databases provide information on glycoforms of glycoproteins. The complex carbohydrate structure database (CCSD)/CarbBank was developed and maintained by the Complex Carbohydrate Research Center of the University of Georgia (USA) [52,53]. It is a reservoir of carbohydrate structures gathered through manual retrospective extraction from existing literature and all the publications in which complex carbohydrate structures were cataloged. Owing to lack of funding, the database has been no longer updated. However, holding 50,000 records from 14,000 publications and 23,500 different carbohydrate sequences, it remains as the largest carbohydrate-related repository. Offshoots of CCSD data have resulted in open access databases, such as GLYCOSCIENCES.de [54], KEGG Glycan database [55], CFG Glycan Database [56], Bacterial Carbohydrate Structure Database (BCSDB) [57] and GlycoBase (currently inactive replaced by GlyStore) [58]. The Japan Consortium for Glycobiology and Glycotechnology DataBase (JCGGDB), which assembles CabosDB, Galaxy, LipidBank, GlycoEpitope, LfDB, and Structural Glycomics Calculation (SGCAL) (1490 carbohydrate structures) and GlycoSuiteDB (3300 unique carbohydrate structures) [59] are freely accessible as well. The EUROCarbDB project aims at developing a new infrastructure, where scientists themselves can upload carbohydrate structure-related data. A prototype that can store carbohydrate structures, has been developed. Primary experimental data (MS, HPLC, and NMR) that serves as reference data for the carbohydrate structure in question can also be uploaded.

Mutual exchange between carbohydrate databases is reported to be a challenging task [60], since various databases use different sequence formats to encode carbohydrate structures [61]. This is why glyco-informatics exists as multiple disconnected and incompatible islands of experimental data, data resources and specific applications, managed by various independent consortia, institutions and research groups [43,62] and no comprehensive database for carbohydrate structures exists. Such isolated islands of information make it difficult for the user to access and derive required information. The difficulty encountered by the users is that they have to visit different database portals to retrieve information on a specific carbohydrate structure and they need to acclimatize themselves with the different local query options, which demand prior knowledge on encoding of the residues in that respective database. In 2005, a comprehensive index of all available structures was initiated by translating the freely available databases to the GlycoCT sequence format [61] and storing in a new database known as GlycomeDB [63]. A web interface was developed as a single query point for all carbohydrate structure databases [64].

A novel bioinformatics method called GlycoMinestruct has been developed. This improves the prediction of human N- and O-linked glycosylation sites by combining sequence and structural features in an integrated computational framework with a two-step feature-selection strategy [65]. *GlycoMinestruct* is established to outperform NGlycPred, which is the only predictor that incorporated both sequence and structure features, achieving high AUC values (in the field of pharmacokinetics, the area under the curve (AUC) defines the variation of a drug concentration in blood plasma as a function of time) for N- and O-linked glycosylation. *GlycoMinestruct* has been used to screen the human proteome and obtain high-confidence predictions for N- and O-linked glycosylation sites and expedited discovery of glycosylation events and substrates facilitating hypothesis-driven experimental studies. There are other glycosylation online predictors available too, which predict N-Glycosylation, O-glycosylation and C-mannosylation sites in the sequence of interest. For example NetNGlyc [66] used for N-Glycosylation site prediction, NetOGlyc [67], YinOYang [68] for O-Glycosylation sites prediction, GPP [69] for N-Glycosylation and O-Glycosylation sites prediction likewise NetCGlyc [70] for C-mannosylation sites prediction) and GlycoMine [71] used for N-, C- and O-linked glycosylation sites prediction. These servers are designed based on different machine learning algorithms like SVM (Support Vector Machine), ANN (Artificial Neural Networks) and RFA (Random Forest Algorithm). OGlycBase [72] is the database of O- and C-glycosylated proteins, contains the experimentally verified O- and C-glycosylated proteins. Whereas the tools like GPI-SOM [73], PredGPI [74] and BIG-P [75] are used for Glycosylphosphatidylinositol modification site (GPI-anchors) prediction. These tools are designed based on SVM and Hidden Markov Model (HMM) algorithms. GlycoFragment and Distance Mapping are also used for glycosylation 3D structure prediction [76].

Bioinformatic tools have also been able to enhance the prediction of glycosylation sites. Glycosylation is able to affect the function of the modified protein and so methods have been developed to predict the glycosylation sites based on the amino acid sequences. The statistical analysis of amino acids surrounding the glycosylation binding sites is dealt with by the being investigated by German Cancer Research Center. One of their tools GlySeq [77], statistically analyzes the amino acids surrounding the glycosylation sites based on protein sequences acquired from Swiss-Prot and Protein Data Bank (PDB), available on the GlySeqDB database. In addition to analyzing the surround sequence, a tool called GlyVicinity performs a statistical analysis of a PDB entry. At Johns Hopkins University, a model to mathematically formulate N-glycosylation was developed [78] whereby, given a set of expressed genes, the list of possible glycans synthesized by the input can be predicted. An enhanced version of this model is currently in use. Recent developments in MS and ionization techniques have enabled the effective structural analysis of glycans [78,79]. The development of glyco-bioinformatics databases and tools such as UniCarb KB [80], GlycoMod [81], GlycoWorkBench [82], GlycReSoft [83] and Multiglycan [84] have accelerated the speed of glycan characterization [28].

Glycosciences DB [85] is a comprehensive data collection linking glycomics and proteomics data, which consists of the databases (http://www.glycosciences.de/db-overview.php) such as GlycoCD-glycoproteins/glycosphingolipids/carbohydrate recognition sites carbohydrate binding lectins/GlycoMapsDB-conformational maps of glycans and MonosaccharideDB-comprehensive resource of these monosaccharides and the tools like Sweet-II, GlyProt, PDB CArbohydrate REsidue check (pdb-care), pdb2linucs, Carbohydrate Ramachandran Plot (CARP), LINUCS, LiGraph, SUgar MOtif search (sumo), extraction of carbohydrate information from pdb-files, glycosidic linkage torsions, glycosylation sites statistics, oligosaccharides 2D structure drawings of which used for display the glycan structures and N-and O-glycan cores motif identification. In 2008, the work group “Frontiers in glycomics” emphasized the need for a curated, sustainably funded glyco-structure database [62,86]. This proposed structural database contains associated information about experimental and biosynthetic data accounting for various analytical methods used to assign glycan structures. Pioneering attempts for a comprehensive glycomic database resulted in Carbank [52] in the 1980s. Although this attempt came to a halt, its assembled data lived on in the next-generation databases including SWEET-DB [87] and GlycosuiteDB [88] (later incorporated into UniCarbKB [89] and GlyConnect [90], having the same agenda as their ancestor. Then, integrative initiatives arose with the goal of centralizing scattered data [e.g., GlycomeDB [63]], and combining it with in silico analytical tools such as GLYCOSCIENCES.de [54], KEGG GLYCAN [55,91] and repositories provided by the Consortium for Functional Glycomics (CFG). The progress in the field was chaotic at the beginning of the century, but progressed considerably through the deployment of a new generation of centralized and integrative resources. These are GlyGen in the US, Glycomics@ExPASy [92] in Europe and GlyCosmos in Japan. GlyCosmos includes GlyTouCan [93], a registry providing glycan structures and develops complementary repositories. Each unique glycan can be correlated with MS data, HPLC retention times and NMR spectra using GlyTouCan.

UniCarb-DB was launched in 2011, to capture information from glycomics MS/MS data, [94,95]. Several versions of UniCarb-DB have been released with the primary aim focusing on improving the quality of glycomics data. UniCarb-DB is now integrated with Glycomics@ExPASy and provides the framework for accessing experimental MS data of biological origin. MS fragmentation glycan spectra are included in the NIST Glycan Mass Spectral Reference Library [96]. Compared to the progress made in glycan structure databases, slow progress has only been made towards the development of softwares for data analysis of glycomic data. GlycosidIQ automated the comparison of observed fragments with theoretical glycofragments from structural databases [97], this automation has been used in a commercial software [98]. GlycoReSoft was used for glycan detection from LC-MS data to compare different samples [99]. Other approaches were able to convert mass spectra into structures [95,100]. More advanced tools for glycomics analysis use partial de novo sequencing [101] including GlycoDeNovo [102] and the recently published Glycoforest [103]. High throughput glycomics MS annotation GRITS Toolbox [104] and quantitation tools [105] are now available.

Entire glycomes of cell systems have been characterized using advanced analytical tools and these data have been stored in glycosylation-specific databases like the GlycomeDB [106] and the Consortium for Functional Glycomics (CFG) website [56]. Bioinformatics and statistical approaches also offer an avenue to study glycosylation [107]. Systems-based mathematical modelling represents an alternate methodology [108,109]. A new MATLAB-based toolbox called Glycosylation Network Analysis Toolbox (GNAT) is able to construct and analyze glycosylation reaction networks. This enables construction, visualization and simulation of glycosylation reaction networks. It enables storing glycan structure information in SBML format files [110] and provides basic manipulation tools to tailor glycans and associated networks. It provides an interactive GUI (Graphical User Interface) that connects individual glycans represented in glycosylation networks and corresponding data in glycomics repositories. Finally, the toolbox facilitates simulation of glycosylation reaction networks [111]. Table 1 presents the overview of the glyco-informatic tools and resources available to date.

## 3. Scarce List of Bioinformatic Resources for Glycosylation in Cancers

It is now known that a plethora of alterations in protein glycosylation can profoundly impact disease and cell phenotypes. The biosynthesis of cancer-associated glycans and the subsequent glycoproteome act synergistically towards disease evolution. It is now proven that there exists a cross talk between glycans and tumor cells [138]. This cross talk leads to the activation of specific oncogenic pathways and brings about invasion and disease dissemination, becoming an important source of relevant glyco(neo)epitopes, holding tremendous potential towards clinical intervention. Glycosylation modifications in cancer include, alterations in glycan length, often toward shorter O-glycans and more branched N-glycans, often marked by distinct changes in glycan sialylation and fucosylation influencing terminal epitopes. Abberant glycosylation has become the fundamental basis for glycan based cancer biomarker discovery, which has been extensively reviewed and reported by many authors [16,139,140,141,142,143,144,145,146,147,148,149,150,151,152,153,154,155,156,157,158,159,160,161,162,163,164,165]. Apart from these, many changes in glycan chains are reported for glycosaminoglycans (GAG). Aberrant glycosylation thus plays a crucial role in tumor progression by regulating tumor proliferation, invasion, metastasis, and angiogenesis [166,167], being frequently cited as a hallmark of cancer [117]. In ovarian cancer, increased branching of the N-glycans attached to glycoproteins [168] and increased sialylation of N-glycopeptides [169] have been confirmed. Because 90% of ovarian cancers are of epithelial origin [170], the glycosylation landscape on cancer cell surface membrane glycoproteins has become diagnostically, prognostically, and therapeutically meaningful [171].

With the increasing evidence of the expression of aberrant glycans being intimately linked with cancer [172], other studies have also shown that altered glycosylation patterns on the surface of cancer cells disrupt normal cellular functions. Lauc et al., 2013 [173] have provided experimental evidence that glycosylation of immunoglobulin G (IgG) influences IgG effector function by modulating binding to Fc receptors. Genetic loci associated with IgG glycosylation were identified using ultra-performance liquid chromatography (UPLC) and MALDI-TOF mass spectrometry (MS). Meta-analysis of genome-wide association study (GWAS) identified 9 genome-wide significant loci strongly associated with various pathological conditions including haematological cancers such as acute lymphoblastic leukaemia, Hodgkin lymphoma, and multiple myeloma. Their study confirmed that it was possible to identify new loci that control glycosylation of a single plasma protein using GWAS in autoimmune diseases and haematological cancers.

Glycosylation is reported to be a global target for androgen control in prostate cancer patients. RNA sequencing analysis of prostate cancer cells and patients by Munkley, 2017 [174] identified 700 androgen-regulated genes. Gene ontology (GO) defined glycosylation as an androgen-regulated process in prostate cancer cells and identified 25 genes with roles in the glycosylation process.

Walsh et al., [48] have described a pipeline and Saldova et al. [175] report the successful identification of biomarkers in breast cancer and ovarian cancer [175] and still others in lung cancer [176] and arthritis [177] using bioinformatics resources. They report an automated LC platform that includes bioinformatics tools such as GlycoBase, GlycoDigest. GlycoMarker, helps identify biomarkers in LC profiles. It is able to statistically test markers, visualize and model algorithms. GlycoMarker was effectively tested on a dataset of 62 breast cancer and 107 control LC profiles [175] and biomarkers for breast cancer were identified. Glycomarker is no longer active.

Breast cancer is the most common cancer in women worldwide, and resistance to the current therapeutics, is an increasing medical challenge. Recently, there is strong evidence pointing out towards the role of glycosylation in tumor formation and metastasis. Kaur et al. [178] have studied the role of sialoglycoproteins in breast cancer by using bioinformatics tools, based on UNIPROT database. By collection of FASTA sequences of breast cancer gene sequence, a database in the form of a word document was constructed. Glycosylation was studied using YinOYang tool on EXPASY, followed by inclusion of differentially expressed genes using KEGG, DAVID and Ingenuity databases. Significant changes in the expression profiling of glycosylation patterns of various proteins associated with breast cancer was also identified. Bioinformatics (involving yinOyang tool on expasy, followed by involvement of differentially expressed genes in important molecular and signaling casades using KEGG, DAVID and Ingenuity databases) worked hand in hand to prove that differential aberrant glycosylated proteins in breast cancer cells are a crucial factor for the overall progression and development of cancer. These were the limited reports that were available involving the use of informatics tools for glycosylation in cancers as summarized in Table 2. As observed from the handful of information in Table 2, this review stress on the fact that more work needs to be done in this area.

## 4. Future Perspective

The transversal nature of glycans and the existence of crosstalk between glycans and stromal components of tumor microenvironments is well established. No the focus is on including glycans and glycoconjugates in comprehensive panomic models, for bringing about progress in precision medicine. Precision medicine is the new age medical approach whereby personalized individual care and medical counsel is given. Glycotope GmbH, founded in 2001, developed GlycoExpress technology [182], which can produce and optimize humanized glycoproteins [183]. They have developed novel technologies for production of biopharmaceuticals based on fully humanized glycoproteins. [182,183]. Thus, the importance and impact of glycosylation on cancer and its diagnosis and therapy are clearly expanding. Glycosylation thus holds promises towards cancer diagnosis, early detection as well as cancer therapy. Bioinformatics is a core component of any pipeline and greatly speed up the storage, annotation and analysis of the data. Bioinformatics programs enable even inexperienced researchers to handle large data sets and performing analysis with ease and efficacy. Bioinformatic applications have been pivotal in enlarging the visions of proteomics in clinical disease detection and management and they would and should in Glycosylation applications too, From what Aoki-Kinoshita [28] reported as three major databases for complex carbohydrates [Glycosciences.de, KEGG GLYCAN, database by the Consortium for Functional Glycomics (CFG)], we have come a long way. This is evidenced by the exhaustive list of tools presented in Table 1. Through reviewing the current scenario in this area, we discovered that many tools and glyco-informatic resources have become inactive, redundant and void. Some tools have merely been developed and demonstrated by a research group and nothing much has been done on it and it has gone inactive. We speculate the reason for this is the gap between tool development and knowledge of existing tools and practical implementation aspects.

As listed in Section 2, numerous bioinformatics resources are available for glycosylation studies, however, this review found a huge vacuum with respect to reports on the use of these glyco-informatic tools for glycosylation studies with respect to cancer. Aberrant glycosylation actively contributes to tumor progression by regulating tumor proliferation, invasion, metastasis, and angiogenesis and could easily benefit from the use of glyco-informatic tools. With a wide range of bioinformatics tools specifically for glycosylation developed and proven for their efficacy, the fact that not even 1% of these available options have been used for glycosylation in cancers is an alarming inference this review aims to focus. This review emphasizes the need to apply the available glyco-informatics options towards cancer for a better cutting-edge technology transfer. The sad fact in science is when we hold the sophistication and yet refrain from applying it towards practical implementation, especially when the practical implementation pertains to human health and welfare.

When cancer has become a rising concern and constant remedies and milestones are being continuously evaluated, given the fact that glycosylation could shed light on early detection and treatment regimens, bioinformatics tools will no doubt be an added asset. Bioinformatics has eased various tedious protocols and will prove handy in this aspect too. Only when the existing tools are utilized and exploited to the full extent, can any improvisations or improvements to suit the requirements for cancer applications be made. The available options can by themselves bring about benchmark progress in glycosylation in cancer detection and therapy. This review puts forth the need for testing the bioinformatics glyco-informatic resources available for cancer research, which may eventually be the key to a crucial fundamental understanding of this complicated disease [184].

## 5. Conclusions

The bioinformatics options available for glycosylation which is a variant of the post-translation modification of proteins have been reviewed and a consolidated list of the tools and resources available have been presented. The lack of application of the existing glyco-infomatic tools for cancer research has been speculated and the need to do so has been emphasized. The major emphasis this review is highlighting is that given all the available bioinformatics resources and developments in glyco-informatics , there is very less that has been applied into the field of cancer biology. This review exposes the huge gap between available bioinformatics tools and technologies and real time applications. This review is expected to bring about an awakening amidst the glyco-informaticians as well as cancer biologists leading to significant bridging of this gap.

## Figures and Tables

**Figure 1 ijms-21-09336-f001:**
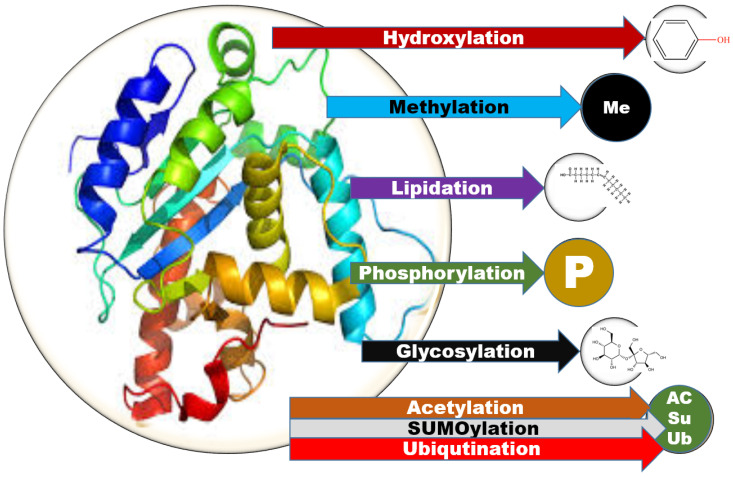
Schematic showing the wide range of post translational modification of proteins of biological significance.

**Figure 2 ijms-21-09336-f002:**
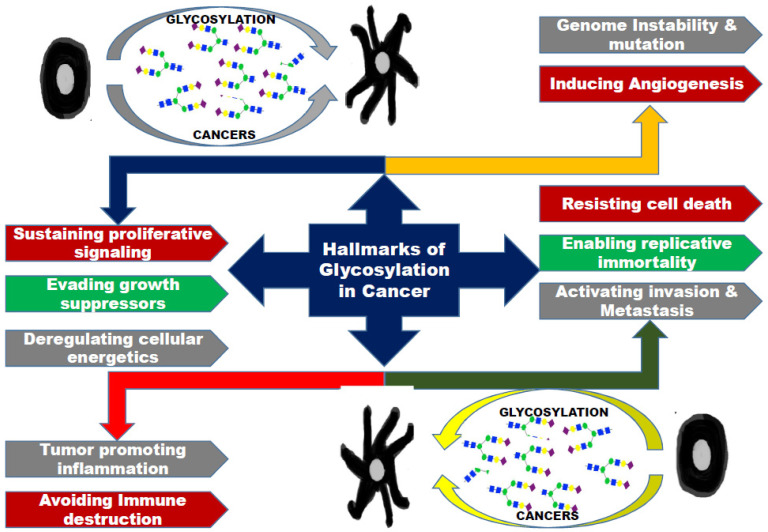
Snapshot of the hallmarks of glycosylation in cancer [15].

**Figure 3 ijms-21-09336-f003:**
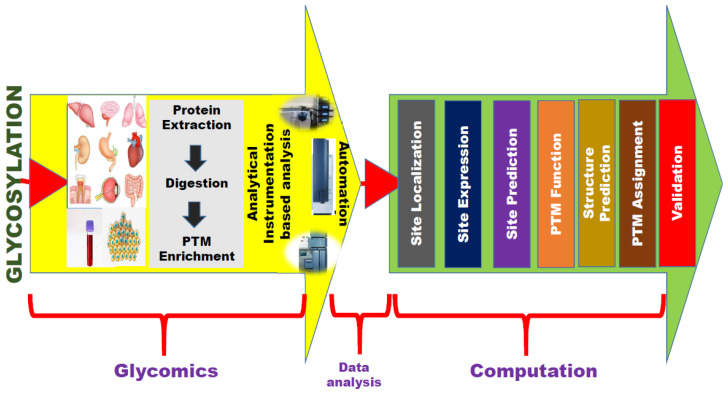
Work-flow of glycosylation research, beginning from wet lab analysis—instrumentational analysis culmination in bioinformatics based computation (data analysis).

**Table 1 ijms-21-09336-t001:** Bioinformatic tool applications in various Glycosylation associated studies.

Bioinformatics Resources	URL	Application	Reference
**Glycosylation Structure Related Resources**			
UniCarb KB	http://unicarbkb.org/	Repository for glycan structures of glycoproteins	[64]
GlycoMod	http://web.expasy.org/glycomod/	Software tool for N-linked and O-linked glycan structures prediction	[84]
*GlycosuiteDB*	http://www.glycosuite.com	Relational database of glycoprotein glycan structures and their biological sources	[88]
*GlycoDeNovo*	https://www.cs.brandeis.edu/~hong/Research/GlycoDeNovo/GlycoDeNovo.htm	Algorithm for Accurate de novo Glycan Topology Reconstruction from Tandem Mass Spectras	[102]
*Glycoforest*	https://glycoforest.expasy.org/	Partial de-novo algorithm for sequencing glycan structures based on MS/MS spectra	[103]
Sweet-II	http://www.glycosciences.de/modeling/sweet2/doc/index.php	Tool to construct 3D models of saccharides from their sequences using standard nomenclature	[112]
GlyProt	http://www.glycosciences.de/modeling/glyprot/php/main.php	Tool to connect N-glycans in silico to a given 3D protein structure.	[113]
PDB CArbohydrate REsidue check (pdb-care)	http://www.glycosciences.de/tools/pdb-care/	Identify and assign carbohydrate structures using atom types and their 3D atom coordinates from PDB-files	[114]
Glyco3D	http://glyco3d.cermav.cnrs.fr/home.php	3D structures of monosaccharides, disaccharides, oligosaccharides, polysaccharides, glycosyltransferases, lectins, monoclonal antibodies against carbohydrates, and glycosaminoglycan-binding proteins	[115]
LiGraph	http://www.glycosciences.de/tools/LiGraph/	Convert a sugar graph to ASCII IUPAC sugar nomenclature or as a graph	[85]
KEGG Glycan database	https://www.genome.jp/kegg/glycan/	Database for glycan structures	[55]
*GlyTouCan*	https://glytoucan.org/	Repository for glycan structures.	[93]
**Resources for Analytical Glycosylation**			
*UniCarb-DB*	https://unicarb-db.expasy.org/	Repository for glycomic MS data	[89]
GlycanBuilder, GlycoWorkBench	http://www.eurocarbdb.org/applications/structure-tools	Tool to annotate Mass Spectra of Glycans	[116]
Multiglycan	https://bio.tools/multiglycan	Information provider for glycan profile information from LC-MS Spectra.	[117]
GlycoFragment and Distance Mapping	www.dkfz.de/spec/projekte/fragments/	Web tool to interpret mass spectra of complex carbohydrates	[76]
**Prediction of Glycosylation**			
NGlycPred	https://bioinformatics.niaid.nih.gov/nglycpred/	Server to predict N-linked Glycosylation sites	[118]
UNIPEP	http://www.unipep.org/	Human and mouse N-glycosylated proteins and their N-glycosylation sites for biomarker discovery	[119]
GlycoEP	https://bio.tools/GlycoEP	prediction of N-linked, O-linked and C-linked glycosites in eukaryotic glycoproteins	[120]
GlySeq	http://www.glycosciences.de/tools/glyseq/	Analyze the sequences around glycosylation sites.	[77]
SPRINT-Gly	https://sparks-lab.org/server/sprint-gly/	predictingN-andO-linked glycosylation sites	[121]
DictyOGlyc	http://www.cbs.dtu.dk/services/DictyOGlyc/	Prediction of O-glycosylation sites in Dictyostelium discoideum proteins	[73]
GlycoMine^struct^	https://glycomine.erc.monash.edu/Lab/GlycoMine_Struct/index.jsp	Highly accurate mapping of the human N-linked and O-linked glycoproteomes	[65]
GlycoForm	http://www.boxer.tcd.ie/gf/	Mathematical model software for N-linked glycosylation	[78]
Glycopep	http://hexose.chem.ku.edu/predictiontable.php	N-linked glycosylation based on target protein and CID spectra analysis	[122]
Consortium for Functional Glycomics (CFC)	http://www.functionalglycomics.org/static/consortium/consortium.shtml	Glycomics resources of glycans and glycan-binding protein	[63]
NetNGlyc	http://www.cbs.dtu.dk/services/NetNGlyc/	N-Glycosylation site predictor	[123]
NetOGlyc	http://www.cbs.dtu.dk/services/NetOGlyc/	O-Glycosylation sites predictor	[75]
GPP	https://comp.chem.nottingham.ac.uk/glyco/	N- and O-Glycosylation site predictor	[73]
Big-PI	https://mendel.imp.ac.at/gpi/gpi_server.html	GPI anchors predictor	[124]
GPI-SOM	http://gpi.unibe.ch/	GPI anchors predictor	[125]
PredGPI	http://gpcr.biocomp.unibo.it/predgpi/	GPI anchors predictor	[85]
NetCGlyc	http://www.cbs.dtu.dk/services/NetCGlyc/	C-mannosylation sites prediction	[123]
**Carbohydrate Knowledgebases**			
GlycoCT	www.eurocarbdb.org	Tool to convert unifying sequence format for carbohydrates	[61]
*SWEET-DB*	http://www.pdg.cnb.uam.es/cursos/Leon2002/pages/software/DatabasesListNAR2002/summary/300.html	Repository for annotated carbohydrates	[87]
SUgar MOtif search (sumo)	http://www.glycosciences.de/tools/sumo/	Tool to search sugar motif regions from carbohydrate structures	[126]
Bacterial Carbohydrate Structure Database (BCSDB, 6789)	http://csdb.glycoscience.ru/bacterial/	Repository for prokaryotic carbohydrate structures, taxonomy, bibliography, NMR data, etc.	[57]
pdb2linucs	http://www.glycosciences.de/tools/pdb2linucs/	Extract carbohydrate information from pdb-files and display it using the LINUCS-Code	[77]
CCSD	https://cordis.europa.eu/project/id/BIOT0184	Information system of carbohydrate science	[52]
Carbohydrate Ramachandran Plot (CARP)	http://www.glycosciences.de/tools/carp/	Analyze carbohydrate data from PDB files using the pdb2linucs algorithm	[127]
LINUCS	http://www.glycosciences.de/tools/linucs/	LInear Notation for Unique description of Carbohydrate Sequences	[128]
**Miscellaneous Resources for Glycosylation**			
RESID	https://proteininformationresource.org/resid/togm.shtml	Table of Glycosylation Modifications	[129]
*GlyGen*	https://www.glygen.org/	Informatics Resources for Glycoscience	[130]
LfDB	https://acgg.asia/lfdb2/	Lectin Frontier DataBase	[131]
CFG	http://www.functionalglycomics.org/	Comprehensive resource for functional glycomics research	[56]
GlycoStore	https://www.glycostore.org	A curated database of information on glycan retention properties with chromatographic, electrophoretic and mass-spectrometry composition data.	[132]
Galaxy, LipidBank	https://jcggdb.jp/database_en.html	Consortium for Glycobiology and Glycobiotechnology database	[133]
Glycosylation Network Analysis Toolbox (GNAT)	http://gnatmatlab.sourceforge.net/	MATLAB-based environment for systems glycobiology	[111]
*Glycomics@ExPASy*	https://www.expasy.org/search/glycomics	Expasy resources for glycomic data.	[134]
MonosaccharideDB	http://www.monosaccharidedb.org/	comprehensive resource of monosaccharides	[135]
GlycReSoft	http://www.bumc.bu.edu/msr/glycresoft/	Software for Glycomics and Glycoproteomics	[99]
CarbBank	https://www.genome.jp/dbget-bin/www_bfind?carbbank	Database management program and the project system of CCSD	[53]
*GlyCosmos*	https://glycosmos.org/	Comprehensive web resource for the glycosciences	[136]
SysPTM	http://lifecenter.sgst.cn/SysPTM/	Resource for post-translational modification	[137]

**Table 2 ijms-21-09336-t002:** Bioinformatic tools for glycosylation in cancer applications.

Bioinformatics Resources	Cancer Type	URL	Application	Reference
RNA sequencing analysis of genes	prostate cancer cell lines and patients	A composite sequencing server	RNA sequencing analysis of identified a set of 700 androgen-regulated genes.	[174]
Gene ontology (GO)	prostate cancer cell lines and patients	Gene ontology (GO) is general functional annotation server	identified 72 terms with significant gene enrichment (*p* < 0.05) and defined glycosylation as an androgen-regulated process in prostate cancer cells.	[174]
GlycoBase (inactive now)	Breast cancer cells	(https://glycobase.nibrt.ie):	database of experimentally determined glycan structures originally developed from the EurocarbDB project	[58]
GlycoDigest	Breast cancer cells	(http://www.glycodigest.org):	a tool that simulates exoglycosidase digestion based on controlled rules acquired from expert knowledge and experimental evidence available in GlycoBase	[179]
GlycoMarker	Breast cancer cells	https://glycobase.nibrt.ie/glycomarker	Web application/server for Biomarker discovery, identifies markers in LC profiles	[48]
using KEGG, DAVID and Ingenuity databases, uniprot database	Breast cancer cells	KEGG, DAVID and Ingenuity databases, uniprot database these are very general functional analysis and sequence databases	Significant change in the expression profiling of glycosylation patterns of various proteins associated with Triple negative breast cancer was identified. Differential aberrant glycosylated proteins in breast cancer cells with respect to non-neoplastic cells	[178]
genome-wide association study (GWAS)	haematological cancers such as acute lymphoblastic leukaemia, Hodgkin lymphoma, and multiple myeloma	GWAS is a general study for genome wide citation	identify new loci that control glycosylation of a single plasma protein using GWAS	[178]
YinOYang	Breast cancer cells	http://www.cbs.dtu.dk/services/YinOYang/	O-Glycosylation sites predictor	[125]
ANCOVA/MANCOVA statistics	Lung Cancer cells	https://www.statisticssolutions.com/multivariate-analysis-of-covariance-mancova/	Statistical analysis	[176]
Student’s *t* test, orthogonal partial least squares discriminant analysis and receiver operating characteristic curve	colorectal cancer tissues (CRC) in Chinese patients	Statistical tools	Statistical analysis of MS data from Linear ion trap quadrupole-electrospray ionization mass spectrometry, on CRC tissues	[180]
Perseus™ (Max Planck Institute of Biochemistry, Berlin, Germany)	Breast cancer	https://pubmed.ncbi.nlm.nih.gov/29344888/	Perseus™ 1.5.2.6 was used for hierarchical clustering, principal component analysis (PCA), and plotting for visualization and statistics	[181]

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
