# Peer review of "Insights into Bioinformatic Applications for Glycosylation: Instigating an Awakening towards Applying Glycoinformatic Resources for Cancer Diagnosis and Therapy"

_ijms, 2020, doi:10.3390/ijms21249336_

Round 1

Reviewer 1 Report

The paper is improved but there is still major concerns and I still feel there is a lack of effort on the glycosylation cancer informatics reviewed.

A.

You replied to me with respect to the small amount of cancer bioinformatics in the work: ‘This review is expected to bring about an awakening amidst the glycoinformaticians as well as cancer biologists’

I think the title is confusing. I think the ‘awakening’ needs to be stronger in the title.

‘Insights into the bioinformatic applications for glycosylation: highlighting the need for more glycoinformatics for cancer diagnosis and therapy’

There is nothing about this ‘awakening’ in the abstract too.

The section ‘Bioinformatic resources for GlycosylationGlycosylation in Cancers’ should read ‘Scarce list of Bioinformatic resources for GlycosylationGlycosylation in Cancers’ to awaken people

‘The use of bioinformatics resources specifically in the area of glycosylationglycosylation in cancers is presented herein.’

This is also not describing the fact that more work needs to be done in this area. I suggest rephrasing it to indicate more work is needed.

B.

‘there is nothing at all applied into the field of cancer biology’ I think ‘nothing’ is too strong there is something but small.

C.

‘Glycosylation modifications in cancer include, alterations in glycan length, often toward shorter O-glycans and more branched N-glycans’

There needs to be an extensive list of citations here. I currently believe there is not enough. I would also suggest a table/figure that shows a list of all glycan biomarkers in all cancers that were found either using bioinformatics or statistical analysis. The latter although not strictly bioinformatics can be considered a form of bioinformatics

For example, I will point you to three papers I randomly found on google scholar:

  • https://pubs.acs.org/doi/abs/10.1021/pr101034t, informatics: 50−50 ANCOVA/MANCOVA statistics

·         https://academic.oup.com/glycob/article-abstract/29/5/372/5304492, partial least squares discriminant analysis

·         https://journals.plos.org/plosone/article?id=10.1371/journal.pone.0231004, Visualization and statistical analyses

·         Lauc et al, 2013 [146], GWAS

·         Munkley, 2017 [147], GO

there are lots of papers on glycan biomarker discovery and they all use different statistical techniques/bioinformatics.

This would be a valuable table. Perhaps, the following columns would be a useful satrt:

[paper], [biomarker], [increased/decreased in cancer], [bioinformatics method used to detect]

Mass spectrometry analysis reveals aberrant N-glycans in colorectal cancer tissues, Diagram of glycan, increased, partial least squares discriminant analysis

More table rows

More table rows

Another example: 'Bioinformatics worked hand in hand to prove that differential aberrant glycosylated proteins in breast cancer cells are a crucial factor for the overall progression and development of cancer'

what type of bioinformatics?

Author Response

The paper is improved but there is still major concerns and I still feel there is a lack of effort on the glycosylation cancer informatics reviewed.

A.

You replied to me with respect to the small amount of cancer bioinformatics in the work: ‘This review is expected to bring about an awakening amidst the glycoinformaticians as well as cancer biologists’ I think the title is confusing. I think the ‘awakening’ needs to be stronger in the title. ‘Insights into the bioinformatic applications for glycosylation: highlighting the need for more glycoinformatics for cancer diagnosis and therapy’

Ans. We would like to thank our dear reviewer for offering your valuable suggestions and moreover working with us towards raising the quality of our publication. We do completely agree with you regarding lack of translating the impression on the awakening theme on the title we have now changed the title to –

Insights into bioinformatic applications for glycosylation: instigating an awakening towards applying glycoinformatic resources for cancer diagnosis and therapy

There is nothing about this ‘awakening’ in the abstract too.

Ans. We have added this theme in the abstract of the revision. Thank you.

The section ‘Bioinformatic resources for GlycosylationGlycosylation in Cancers’ should read ‘Scarce list of Bioinformatic resources for GlycosylationGlycosylation in Cancers’ to awaken people

 Ans. Yes, very aptly proposed, we have now revised the subheading just as you have suggested. Thanks a lot.

‘The use of bioinformatics resources specifically in the area of glycosylationglycosylation in cancers is presented herein.’

This is also not describing the fact that more work needs to be done in this area. I suggest rephrasing it to indicate more work is needed.

Ans. Yes you are right, we have now removed the sentence in the revision.

B.

‘there is nothing at all applied into the field of cancer biology’ I think ‘nothing’ is too strong there is something but small.

 Ans. Thank you, we have repharased that sentence as you have suggested.

C.

‘Glycosylation modifications in cancer include, alterations in glycan

length, often toward shorter O-glycans and more branched N-glycans’

There needs to be an extensive list of citations here. I currently believe there is not enough. I would also suggest a table/figure that shows a list of all glycan biomarkers in all cancers that were found either using bioinformatics or statistical analysis. The latter although not strictly bioinformatics can be considered a form of bioinformatics For example, I will point you to three papers I randomly found on google scholar:

  • https://pubs.acs.org/doi/abs/10.1021/pr101034t, informatics: 50−50 ANCOVA/MANCOVA statistics
  • https://academic.oup.com/glycob/article-abstract/29/5/372/5304492, partial least squares discriminant analysis
  • https://journals.plos.org/plosone/article?id=10.1371/journal.pone.0231004, Visualization and statistical analyses
  • Lauc et al, 2013 [146], GWAS
  • Munkley, 2017 [147], GO

there are lots of papers on glycan biomarker discovery and they all use different statistical techniques/bioinformatics.This would be a valuable table. Perhaps, the following columns would be a useful satrt:[paper], [biomarker], [increased/decreased in cancer], [bioinformatics method used to detect]Mass spectrometry analysis reveals aberrant N-glycans in colorectal cancer tissues, Diagram of glycan, increased, partial least squares discriminant analysisMore table rowsMore table rows

Ans. We have added citations. We would like to explain here, that the reason we did not tabulate the general list of glycan cancer biomarkers, is because there are almost 3 to 4 excellent reviews with elaborate tables exclusively on cancer biomarkers and glycosylation (Glycosylation-Based Serum Biomarkers for Cancer, Alan Kirwan,1 Marta Utratna,1 Michael E. O’Dwyer,2 Lokesh Joshi,1 and Michelle Kilcoyne; Franca Maria Tuccillo, Annamaria de Laurentiis, Camillo Palmieri, Giuseppe Fiume, Patrizia Bonelli, Antonella Borrelli, Pierfrancesco Tassone, Iris Scala, Franco Maria Buonaguro, Ileana Quinto, Giuseppe Scala, "Aberrant Glycosylation as Biomarker for Cancer: Focus on CD43", BioMed Research International, vol. 2014, Article ID 742831, 13 pages, 2014. https://doi.org/10.1155/2014/742831; Kailemia MJ, Park D, Lebrilla CB. Glycans and glycoproteins as specific biomarkers for cancer. Anal Bioanal Chem. 2017;409(2):395-410. doi:10.1007/s00216-016-9880-6; Alterations in glycosylation as biomarkers for cancer detection Celso A Reis1,2, Hugo Osorio1, Luisa Silva1, Catarina Gomes1, Leonor David; Protein Glycosylation and Tumor Microenvironment Alterations Driving Cancer Hallmarks, Andreia Peixoto1,2,3,4, Marta Relvas-Santos1, Rita Azevedo1,2, Lúcio Lara Santos1,5 and José Alexandre Ferreira; Wang, M., Zhu, J., Lubman, D. M., & Gao, C. (2019). Aberrant glycosylation and cancer biomarker discovery: a promising and thorny journey, Clinical Chemistry and Laboratory Medicine (CCLM), 57(4), 407-416. doi: https://doi.org/10.1515/cclm-2018-0379. With these excellent reviews driving home all the biomarker discovery information with respect to glycosylation, and as it is with this review already asked to reduce content by the reviewers, we at this moment have not added a table on biomarkers. And with respect to cancer biomarker discovery and bioinformatics inputs almost nothing is there.

However, we have added rows in Table 2 with respect to the statistical analysis packages in the area of cancer as per your suggestion.  Thank you for your kind understanding.

 example: 'Bioinformatics worked hand in hand to prove that differential aberrant glycosylated proteins in breast cancer cells are a crucial factor for the overall progression and development of cancer' what type of bioinformatics?

Ans. We have added more information on this aspect, in the revised text. We have elaborated on this in the appropriate place in the revision. Thank you very much

Reviewer 2 Report

The authors have addressed the major concerns raised by previous review.  Other than a few minor spelling/punctuation issues that should be corrected prior to going to press, the article is essentially ready for publication.

Author Response

Thankyou very much for your meticulous review, we have worked on the language issues, thank you again

Reviewer 3 Report

The authors have addressed most of the reviewer concerns.

Author Response

Thankyou very much, we appreciate your efforts and inputs

Round 2

Reviewer 1 Report

The two revisions i requested have addressed all my concerns

This manuscript is a resubmission of an earlier submission. The following is a list of the peer review reports and author responses from that submission.

Round 1

Reviewer 1 Report

The authors review important bioinformatics tools related with glycomic analysis. Since these approaches are quite complex, this work will greatly benefit from the inclusion of an in-depth tutorial explaining, with an example, the proper usage of a combination of the complementary tools described in this review.

Reviewer 2 Report

Overall, this review is important as it draws attention to a significant but often overlooked phenomenon-- glycosylation.  In large part this disregard by the general scientific community is due to the complexity and variety of glycans which are synthesized via non-template-based mechanisms; simply put, glycosylation is not as easy to predict or characterize as nucleic acid or protein synthesis.  Therefore, this review's focus on the variety of bioinformatic tools now available to study glycosylation and yet the apparent neglect of these tools for practical applications such as cancer research is quite timely.  

Despite the importance of this topic, enthusiasm for the paper is dampened considerably by a variety of language and style issues as well as several citation errors.  In general, there are multiple punctuation problems, awkward sentence structure, and incomplete sentences or fragment statements peppered throughout the manuscript.  In short, the manuscript requires significant copy editing by someone who is an expert in the English language. 

The following are specific alterations that are needed to improve the manuscript:

(1) In the last paragraph of the Introduction section: The first sentence should stress that this is the first review to look at the use of glycosylation bioinformatics data in cancer research and diagnostics.

(2) On page 5: Eliminate the first sentence of the paragraph that starts as, "Glycosylation plays an important role in cell-cell adhesion, ligand-binding, and subcellular recognition." Though true, this is not the topic of the paragraph and therefore is out of place.  The paragraph is primarily discussing protein glycosylation site predictions.  Moreover, this sentence is redundant as it is essentially mentioned in the Introductino.

(3) On page 6: Eliminate the first sentence of the paragraph that begins, "Glycosylation can alter a majority of mammalian-secreted and cell surface proteins,..." as it, too, doesn't match the topic of the paragraph.  In addition, it is a poorly written sentence.

(4) On page 5: Define "AUC values."

(5) The word "glycosylation" should not be capitalized unless it is the first word of the sentence.

(6) In Section 2 ("Snapshot of bioinformatic resources for glycosylation"): The last 2 paragraphs should be combined with the 4th paragraph that already introduces predicting protein glycosylation.  Since there is some redundancy, these 3 paragraphs could probably be reduced down to 1 or 2 paragraphs.

(7) On page 7: The last sentence of the first paragraph is much too long and confusing.

(8) Section 3 should clearly state that there is a lack of glycoinformatic tools currently being used in cancer research.  Authors should not wait until "Future Perspective" section to discuss this dilemma. 

(9) In Section 4 ("Future Perspective"): Change "a-2,6-linked sialylation" to "alpha-2,6-linked sialylation."

(10) In Section 4 on page 14: The last sentence in the first paragraph needs work as it is long and confusing.

(11) In Section 4: Replace the word "upgradation" with "improvements."

(12) In Section 4, last sentence: Replace the phrase "unresolvable disease" with "complicated disease."

(13) The Conclusion should be written in a more provocative or exciting manner.

(14)  Table 1: Why isn't the table organized alphabetically by database?

(15) Table 1: Neither the Cancer Genome Atlas database (which is curiously listed TWICE!) nor GISTIC2.0 should be included as neither have anything to do with glycosylation.

(16) Table 2 needs a lot of work.  It is unclear from several of the entries what each column is for-- need to label the columns and stick with that format.  Will also need to eliminate the first entry, as the phrase "I couldn't find this" is not exactly professional. Finally, it is unclear why one entry is in BOLD.

(17) For reference 7, the authors should consider replacing with the following review article that better describes the general effects of glycosylation (specifically polysialylation) on cell-cell interactions-- Troy, FA, 1992, Polysialylation: from bacteria to brains, Glycobiology 2(1): 5-23.

(18) Authors should consider adding in a few citations re: polysialic acid and cancer.

(19) Reference 22 is cited in the text as a study on colorectal cancer, but it is actually on testicular cancer.  This must be corrected.

(20) Reference 102 is cited as a GlycoMod reference, but it seems that the true GlycoMod reference is 105.

(21) Reference 104 is cited as a GlycReSoft reference, but it appears to be a GlycoPep ID reference instead.

(22) Reference 101 is a GLYCO-FRAGMENT reference, not UniCarb KB as stated in the text and Table 1.

(23) In Section 3, first paragraph: Is reference 115 truly the best "hallmark of cancer" reference?

(24) In Section 3, second paragraph: Gordaner et al. is cited for reference 146, but the author is actually Lauc et al.

(25) The references for Table 1 need to be rechecked.  At least 3 errors were found, and there may be more.

(26) Reference 87: This was In preparation in 2004.  It should have been published by now!  Find the published version.

Reviewer 3 Report

The paper overall is readable. However, the paper seems to rehash older review papers and only ever really focuses on glycoinformatics for cancer in section three which is very short and mainly focuses on breast cancer only.  My biggest criticism is that many of the software cited are inactive and have broken links making them unimportant for glycosylation informatics today. To repeat: (1) there is very little on cancer glycoinformatics specifically and (2) many of the software cited is broken.

In the following ‘’xxx’ is a direct quote from the paper. The following are MAJOR concerns:

Abstract

  1. I suggest removing ‘The MIT’s magazine Technology Review,2003 has identified glycomics’ it is too old for a current review.
  2. ‘The direct implication of glycosylation as a hallmark of cancer and for cancer therapy is well established.’ I think you should state one or two roles here as the ‘well established’ needs a citation which you cannot do in an abstract.
  3. ‘improvised its practical implication’ is improvised the correct word to use here? It seems misplaced to me.

Introduction

  1. ‘Initially, carbohydrates are indirectly encoded in the genome. Both the sequence and structure of glycans are influenced by glycosyltransferases, carbohydrate-modifying enzymes and glycosidases, that affect the glycosidic bonds [12].’ These sentences are a little confusing to the reader only learning about glycosylation. I think what you are trying to write is that the glycotransferases are encoded in the genome. Please rephrase to make this clearer.
  2. ‘This knowledge is useful in improving the 3D protein structure prediction, to manipulate protein pharmacokinetic properties through glycoengineering.’ I did not know this please cite relevant literature.
  3. Figure 2 must have a citation for every hallmark. I do not see a direct correspondence between each hallmark (e.g. avoiding immune destruction) and a citation. Perhaps there is but I don’t see directly. Perhaps you need a table of citations to go along side Figure 2.
  4. ‘The application of bioinformatic tools in biology has led to a fundamental understanding of the biological processes involved.’ This in an opinion. Is there any evidence that they have? Otherwise I think opinions without literature should not be in the article.
  5. ‘Bioinformatics is an interdisciplinary field of life sciences involving:…’. This reads like a popular science article. I feel that bioinformatics is well established now. It doesn’t need to be explained.
  6. ‘Databases are playing a significant role in modern life science, bioinformatics provides databases and tools to support glycobiologists in their research.’ Databases have always played significant roles. Again I don’t think this needs to be explained.

‘Snapshot of Bioinformatic resources for Glycosylation’

  1. ‘essentially handy’ this is very colloquial language not everyone will understand it especially when English is a second language.
  2. ‘Automation technology when combined with bioinformatics leads to several orders of magnitude higher high throughput processing.’ This is an opinion please provide literature.
  3. http://www.eurocarbdb.org in INactive.. see General statements below. It is very important that all resources are checked for inactivity.
  4. ‘It is known that direct cross-linking between the established carbohydrate databases has been difficult’ this has been repeated in many reviews before. Can the authors try not to repeat what is in previous glycoinformatics reviews?
  5. ‘A novel bioinformatics method called GlycoMinestruct’ I think there needs to be a special section devoted to prediction of N- and O- glycosylation sites. The flow of section 2 is very hard to read. For example, when GlycoMinestruct (site prediction) is described, directly after it there is a section on databases, then directly after the databases/glycan sequencing then authors go back to describe site prediction starting with sentence ‘There are many glycosylation online predictors available’
  6. Another example of poor flow starts at setence ‘There are many glycosylation online predictors available...’ then the sentence starting ‘Glycosciences.DB [97] is a comprehensive data collection..’ starts to describe databases and tools which are very little to do with amino acid site prediction. Then, the authors are back writing about site prediction starting at sentence ‘Bioinformatic tools have also been able to enhance prediction of glycosylation sites.’

‘Bioinformatic resources for Glycosylation in Cancers’

  1. This section should be the focus of the paper. The previous section is just a repeat of the other glycoinformatics reviews.
  2. ‘Walsh et al, [43] report the successful identification of biomarkers in breast cancer [148]’ – Walsh et al. only described a pipeline they did not discover the breast cancer biomarkers. This was the work of the reference [148]. As mentioned below glycomarker is inactive. It seems the links are taken from the original papers without checking if they were active.
  3. ‘GlycoMarker, is a tool to easily identify biomarkers in LC profiles. It is a client–server application available to the public over the web (https://glycobase.nibrt.ie/glycomarker).’ And ‘The three main components of GlycoMarker are tools to execute (i) statistical testing of markers, (ii) informative visualization and (iii) modelling algorithms borrowed from statistics and machine learning.’ This seems to be directly copied and pasted from reference 43. Be careful.
  4. ‘potent biomarkers’ why potent? This suggests some form of strength in the biomarker.
  5. GlycoProfileAssigner is not used to find cancer biomarkers rather it is used to quantitate and identify glycans in LC profiles.
  6. Gene Ontology can be used for many bioinformatics applications. Glycosylation cancer would only be a tiny fraction of its use. Therefore, is it really glycoinformatics software for cancer.

General statements:

  1. Additionally, you need to make a very clear case why your paper is different to the previous reviews i.e. it is specifically for helping understand bioinformatics in cancer.
  2. Glycomarker is inactive. This needs to be removed from the manuscript however the report by Walsh et al. is still valid. Please make sure that all software at the time of writing are active. Tables should only contain active software and databases. A supplementary material can contain inactive ones. IMPORTANT: Citing inactive software and databases is a major mistake and should be rectified in this paper if it is to be respected by the community. I did not exhaustively test the software mentioned in the paper if they were inactive. But I tried three http://www.glycosuite.com/ and http://sparkslab.org/server/SPRINT-Gly/ and http://lifecenter.sgst.cn/SysPTM/ and could not link
  3. ‘Owing to lack of funding, the database has been no longer updated.’ Again please do not cite old and inactive software. The paper is already quite long and these types of old software belong in history books not 2020 review articles.
  4. The Figures need better graphic design. They seem to be powerpoint lecture slides at the moment.
  5. Many relevant literature is missed. For example, GlycoStore (https://pubmed.ncbi.nlm.nih.gov/29897488/) has replaced GlycoBase as the latest UPLC database.
  6. The paper is extremely long. Can it be shortened?
  7. One of my bigger critiques about this paper is that you want to explain bioinformatics tools in relation to glycans/glycosylation AND cancer. However, it seems you only do this briefly while the rest of the paper has been described in other reviews. The paper seems to be just a rehash of all the other glycoinformatics reviews that have a general scope.

MINOR:

  1. ‘Munkley et al’ you should cite ref 15 at the first mention of of Munkley
  2. ‘metabolism [15]. Development and progression of cancer is known to result in fundamental changes to the glycome [15]. Aberrant glycosylation has been reported in prostate cancer [15], where aberrations in glycan composition of prostate cancer cells are linked to disease progression [15].’ Ref 15 is over cited here.
  3. [27-31] citation list is quite small given the amount of glycoinformatics reviews in the last 10 years. I also point the authors to:
    1. https://www.sciencedirect.com/science/article/abs/pii/S0959440X19301356
    2. https://pubmed.ncbi.nlm.nih.gov/27522273/
  4. Do not use hyperlinks in the paper. It is better to put the links in the Tables.
  5. ‘Howbeit’ never seen this word before is this old English?
  6. ‘available till date’ perhaps should be xxavailable to datexx